# Evaluating adsorption isotherm models for determining the partitioning of ammonium between soil and soil-pore water in environmental soil samples

Matthew G. Davis*, Kevin Yan, Jennifer G. Murphy

Department of Chemistry, University of Toronto, Toronto, M5S 3H6, Canada

*Correspondence to*: Matthew G. Davis (mg.davis@mail.utoronto.ca)

**Abstract:** Ammonium in soil pore water is thought to participate in bidirectional exchange with the atmosphere; however, common soil nutrient analysis methods determine the bulk quantity of ammonium associated with the soil particles, rather than determining the aqueous ammonium concentration. Previous works have applied the Langmuir and Freundlich isotherm

equations to ammonium-enriched soils to estimate partitioning, but this may not be representative of conditions in natural, unmanaged soils. In this work, environmental soil samples were collected from greenspaces in Toronto and used to evaluate several commonly used adsorption isotherm equations, including the Langmuir, Freundlich, Temkin and Toth equations, to determine their applicability in lightly managed and non-fertilized soils. We then compare ammonia emission potentials (a quantity predicting the propensity of ammonia to volatilize from a liquid reservoir) determined using a conventional high-salt

extraction procedure to determine the soil ammonium content to that modelled using the Temkin and Langmuir equations, and demonstrate that conventional approaches may overestimate emission potentials from soils by a factor of 5 – 20.

Key words: ammonium soil adsorption; emission potential;

## 1. Introduction:

### 1.1. Contextualizing the significance of ammonium partitioning in soils

Globally, emissions of reduced nitrogen compounds ($NH_3$) make up as much as half of the global atmospheric reactive nitrogen sources (Flechard et al., 2013). Of the $NH_3$ budget, approximately two thirds of emissions are related to anthropogenic agricultural activities, with natural sources being responsible for only 15 – 20% of emissions (Bouwman et al., 1997; Sutton et al., 2008, 2013; Van Aardenne et al., 2001). Following emission, $NH_3$ tends to partition into fine particulate matter or deposit via wet or dry deposition on a timescale of hours to days. $NH_3$ is understood to engage in bidirectional exchange throughout

the earth system, with $NH_3$ depositing, or volatilizing depending on local environmental conditions (Farquhar et al., 1980; Flechard et al., 1999, 2013; Guo et al., 2022; Nemitz et al., 2000; Sutton et al., 1995; Walker et al., 2023; Wentworth et al., 2014; Zhang et al., 2010). The bidirectional exchange of $NH_3$ between soils and the atmosphere has been considered important to its overall budget, particularly in remote areas, but research on the mechanisms of $NH_3$ partitioning in soils between adsorbed

(inaccessible) and aqueous (accessible) $NH_3$ often focuses on fertilized croplands with substantial concentrations of $NH_3$

present (Venterea et al., 2015; Vogeler et al., 2011). Consequently, $NH_3$ volatilization models may parameterize all or most of the soil $NH_3$ as being readily able to exchange with the atmosphere, which may be reasonable for recently fertilized soils, but not for unmanaged soils (Massad et al., 2010; Pleim et al., 2013; Zhu et al., 2015). However, cropland is estimated to make up less than 15% of the Earth's land area, while unfertilized or irregularly fertilized natural, semi-natural or pastoral land reflects nearly three quarters of terrestrial surfaces (Ellis et al., 2020). The short atmospheric lifetime of $NH_3$ makes it important to

understand the exchange of $NH_3$ over all types of surfaces, despite agricultural cropland being the most globally significant source.

### 1.2. Importance of developing an ammonium adsorption partitioning model

The propensity for ammonia to volatilize from liquid reservoirs is parameterized by the emission potential ($\Gamma$), which is calculated as the ratio of aqueous $NH_4^+$ to $H^+$. However, $NH_4^+$ in soils is not only present as $NH_{4(aq)}^+$ in soil pore water, but

also present as $NH_{4(ads)}^+$ adsorbed to soil particles. Because soils tend to exist with both permanent and pH-dependent negative charges (Bache, 1976), $NH_4^+$ ions compete for adsorption against other cations in solution. The total quantity of cations that can adsorb to the soil particles is termed the cation exchange capacity (CEC, typically reported as centimoles positive charge/kg). Traditional methods for determining $NH_4^+$ in soil—intended for nutrient analysis in which the total ammoniacal nitrogen is more important than the partitioning between $NH_{4(ads)}^+$ and $NH_{4(aq)}^+$—use high concentration salt solutions to

displace all cations from the soil. As a result, these approaches do not distinguish between $NH_{4(aq)}^+$ and $NH_{4(ads)}^+$ (Li et al., 2012) and therefore likely overestimate the emission potential of soils. In this manuscript, we explore a variety of adsorption isotherm models with the goal of identifying a simple approach to relate the total quantity of $NH_4^+$ in soil to the aqueous fraction of $NH_{4(aq)}^+$ that can participate in bidirectional exchange with the atmosphere.

### 1.3. Adsorption isotherm equations

The adsorption behavior of molecules to surfaces is complex and dependent on the properties of both the surface and the adsorbed molecules. Numerous adsorption equations have been proposed, based on both theoretical and empirical models; however, as the partitioning of $NH_4^+$ between soil pore water and soil particles is complex and influenced by many other simultaneous equilibria, we consider each of the examined equations as being empirically determined only, rather than based on a theoretical treatment of the system. Previous studies have sought to develop an adsorption model for $NH_4^+$ in soils based

on the Langmuir (Alnsour, 2020; Venterea et al., 2015) and Freundlich equations (Vogeler et al., 2011), but have focused on croplands and other agricultural soils. In this work, in addition to the Langmuir and Freundlich equations, we also investigate the application of the Temkin and Toth adsorption equations to $NH_4^+$ sorption in soils. Each of these equations (as formulated in Table 1) represents the adsorbed $NH_4^+$ concentration (S, mg kg$^{-1}$) in terms of the $NH_4^+$ concentration in solution (C, mg L$^{-1}$). Except for the Freundlich equation, each of these equations incorporate a saturation point or maximum adsorption capacity

$(S_{max}$, mg kg$^{-1}$), this work treats $S_{max}$ as an empirically measured property, equivalent to the CEC (converted to mg of NH$_4^+$ kg$^{-1}$ soil), rather than as a calculated fitting parameter.

**Table 1: Adsorption isotherm equations applied in this work and their parameters**

| Adsorption Isotherm Model | Equation[†] | Units | Reference |
|---|---|---|---|
| Langmuir | (1) $S = \frac{S_{max}k_LC}{1+k_LC}$ | $S_{max}$ (mg kg$^{-1}$), $K_L$ (L mg$^{-1}$) | (Langmuir, 1916) |
| Freundlich | (2) $S = k_FC^{n_F}$ | $k_F$, $n_F$ (dimensionless) | (Freundlich, 1909) |
| Temkin* | (3) $S = q_Tln(1 + K_TC)$ | $q_T$ (product of $S_{max}$ (mg kg$^{-1}$) and $\frac{RT}{b}$ (dimensionless)), $K_T$ (L mg$^{-1}$) | (Temkin & Pyzhev, 1940) |
| Toth | (4) $S = \frac{bC}{(K_{To}+C^{n_T})^{\frac{1}{n_T}}}$ | b (product of $S_{max}$ (mg kg$^{-1}$) and a dimensionless scaling factor), $K_{To}$ (mg L$^{-1}$), $n_T$ (dimensionless) | (Tóth, 1995) |

*The Temkin model is also given as $S = \frac{RT}{b}lnK_TC$, see (Chu, 2021) for this formulation

[†] Where S is the concentration of NH$_4^+$ adsorbed to soil particles (mg kg$^{-1}$), C is the concentration of NH$_4^+$ in solution (mg L$^{-1}$), and Smax is the maximum adsorption capacity (mg kg$^{-1}$), which was determined empirically as equivalent to the measured cation exchange capacity of the soil. This work treats the remaining parameters as empirical constants only.

## 2. Methods:

### 2.1. Soil Collection:

Soil samples were collected from 24 greenspace (parks, urban forest, roadside-sites, etc.) locations across Toronto. Samples
were collected only on days preceded by at least two days without precipitation and were collected by inserting a 7.5 cm internal diameter steel tube into the ground to a depth of 5 – 10 cm and recovering a soil core by removing the tube from the ground. The soil cores were mixed, and sieved immediately, and transported back to the lab for analysis. Samples analyzed for NH$_4^+$ content were always analyzed immediately to avoid potential artefacts from freezing, samples analyzed for CEC or adsorption curves were frozen for storage prior to analysis. Soil samples were collected from eight locations in Fall 2021 as a
training set for developing the model. A subsequent 16 soil samples from across Toronto were collected in Spring/Summer 2023 to be used to evaluate the effectiveness of applying the model to uncharacterized soils, and to determine the impact on soil emission potentials.

### 2.2. Soil analysis:

#### 2.2.1. Cation Exchange Capacity determination

The CEC was determined using the inductively-coupled plasma optical emission spectroscopy (ICP-OES) cation sum method (Bache, 1976). Briefly, 1 g of soil was measured out and mixed with 25 mL of 1 M NH$_4$Cl, shaken, and refrigerated for 36 - 48 hours to settle. The supernatant was filtered using 0.2 μm syringe filters, diluted 50-fold using volumetric glassware and analyzed for the common soil-associated exchangeable cations, Na$^+$, K$^+$, Mg$^{2+}$, and Ca$^{2+}$. The ICP-OES (iCAP Pro,

ThermoFisher Scientific, Waltham, USA) was calibrated using a commercially available mixed standard of 6 cations ($Li^+$,

Na$^+$, $NH_4^+$, $K^+$, $Mg^{2+}$, $Ca^{2+}$) (Dionex Cation II, ThermoFisher Scientific, Weltham, USA). Another soil-associated cation that can contribute to CEC is $Al^{3+}$, our initial measurements screened for Al, but we did not detect it in solution. The CEC was determined for all 24 collected samples and used to select three of the soil samples from the 16 collected in 2023 to be used as a test set for the developed model.

### 2.2.2. Adsorption curve characterization

The determination of the adsorption behavior of $NH_4^+$ was performed using a modified version of the procedure described by Venterea et al. (2015), combined with the ICP-OES cation sum method (Bache, 1976). Briefly, a series of batch equilibrium samples were prepared by mixing 1 g of soil with 25 mL aliquots of $NH_4Cl$ solutions with concentrations ranging from 2.5 – 1000 mM. The samples were shaken, refrigerated for 36 – 48 hours to settle, filtered with 0.2 µm syringe filters, diluted 50-fold with volumetric glassware, and analyzed using ICP-OES. The quantity of $NH_4^+$ adsorbed onto each soil was inferred based on the displaced $Na^+$, $Mg^{2+}$, $Ca^{2+}$ and $K^+$ ions measured in solution. In the cation sum method, the total quantity of adsorbable cations is determined by saturating the soil with an index cation (in this procedure, and generally, $NH_4^+$ from $NH_4Cl$), which displaces the exchangeable cations on the soil's adsorption sites. Thus, the displaced cations measured for each $NH_4Cl$ solution concentration are representative of the quantity of $NH_4^+$ adsorbing onto the soil. Adsorption curves were determined for the original eight soil samples in the training set, as well as the three selected soil samples for the test set.

### 2.2.3. pH and $NH_4^+$ determination

pH and $NH_4^+$ content was determined for the 16 soil samples collected in 2023. pH was determined for each soil by mixing soil with ultra-pure (18 MΩ cm) water (DIW) in a 1:2 ratio, and then measuring the pH of the slurry by immersing a pH electrode (Hach Company, Loveland, USA) until a stable pH reading was obtained. $NH_4^+$ was extracted from the soil using a 2 M KCl extraction solution. 2.5 g of soil was mixed with 25 mL of the extraction solution, shaken, and refrigerated for 36 hours to allow suspended solids to settle out of solution. Afterward, the supernatant was filtered using 0.2 µm PES membrane syringe filters. The soil $NH_4^+$ was quantified using the indophenol-blue salicylate method (Kempers & Zweers, 1986). Briefly, two reagent solutions, A and B were prepared: Reagent A consisted of a solution of 1 M sodium salicylate and 100 mg L$^{-1}$ sodium nitroprusside, while Reagent B consisted of a solution of 1 M NaOH and 0.12% by volume of 5% available chloride NaOCl. Soil extracts were prepared for analysis by adding 0.6 mL of reagent A to 2 mL of soil extract, followed by the addition of 1.4 mL of reagent B. The mixtures were then stored for 2 hours in the dark for color development, and then quantified using UV-VIS spectrometry (Lambda 365, Perkin-Elmer, Waltham, USA) at 649 nm.

Conventionally, the soil emission potential is calculated as a function of the soil pH ($H^+_{DIW}$) and salt solution-extracted $NH_4^+$ ($NH_4^+_{SALT}$). In addition to investigating whether adsorption isotherm equations could be applied to estimate $NH_4^+_{(aq)}$ from the total soil $NH_4^+$, we investigated the impact of calculating the emission potential using a 'like-with-like' ratio of $H^+_{DIW}$ with $NH_4^+$ extracted using DIW as the solvent, as well as the ratio of $NH_4^+_{SALT}$ to $H^+$ determined from a salt-solution:soil

slurry. Consequently, the pH was also determined as described above, but using a 0.01 M CaCl$_2$ solution in place of DIW, while soil NH$_4^+$ was also determined as described above, but using DIW as the solvent.

### 2.3. Data analysis

#### 2.3.1. Characterizing adsorption parameters using a training and a test set

Curve-fitting was done in R using the nls function to fit our experimental data for the eight samples in our training set to the Langmuir, Freundlich, Temkin, and Toth equations. Goodness of fit was evaluated by calculating the Akaike information criterion (AIC) using the AIC function from the R *stats* package. The AIC is calculated using equation (5):

(5) AIC = 2K – 2ln(L)

Where K is the number of independent variables, and L is the log-likelihood estimate. The log-likelihood estimate can be extracted directly from nls objects fitted in R using the AIC function.

Fitting parameters for each equation were determined by pooling all the experimental data, standardizing each adsorption curve by the maximum adsorption achieved (i.e., all curves went from 0 to 1), and then fitting each equation from Table 1 to those curves.

To validate the effectiveness of these equations when applied to uncharacterized soil samples, we selected three soil samples from the 16 soil samples collected in 2023 to form a test set, these samples were selected by choosing the soil samples with the lowest (10.9) and highest CECs (37.2), and a soil with an average CEC (25.3). As the original training set mostly consisted of samples with CECs from 20 – 30 (with two samples with CECs of 7.6 and 16), we chose two samples that were significantly different than the average training set sample, as well as one similar sample to determine whether the fitting parameters could be used for 'extreme' samples, or only for samples similar to the training set. The test set was characterized in the same way as the training set, and was then modelled using three approaches:

i.   Using the average CEC for all the soil samples of 25 cmole kg$^{-1}$ (S$_{max}$ of 4500 mg kg$^{-1}$), and the training set parameters.

ii.  Using the measured CEC for each soil sample to calculate S$_{max}$, and the training set parameters.

iii. Using the fitting algorithm as described in Sect.2.3.1 to determine the least squares fit for each equation to the experimental data.

#### 2.3.2. Emission potential determination

The NH$_3$ emission potential is a quantity calculated as the ratio of aqueous NH$_4^+$ to H$^+$ (Eq.(6)).

(6) $\Gamma = \frac{[NH_4^+]}{[H^+]}$

Commonly, for soils this would be calculated using the total NH$_4^+$ (determined using a salt solution extraction) and the pH measured using an extraction with deionized water. We denote this as $\Gamma_{STD}$, corresponding to $\frac{[NH_4^+]_{salt}}{[H^+]_{DIW}}$. The pH may also be

measured in a (less concentrated) salt solution, which we denote as $\Gamma_{SALT}$, corresponding to $\frac{[NH_4^+]_{salt}}{[H^+]_{salt}}$. Similarly, though we are not aware of this as a common method, $NH_4^+$ could be determined using a DIW extraction solution, resulting in a third parameterization of the emission potential as $\Gamma_{DIW}$, representing $\frac{[NH_4^+]_{DIW}}{[H^+]_{DIW}}$. Lastly, by applying one of the adsorption isotherm models, the total soil $NH_4^+$ can be partitioned into $NH_4^+{}_{(ads)}$ (S) and $NH_4^+{}_{(aq)}$ (C), and the emission potential can be calculated

using only the $NH_4^+$ in solution (C). These versions of the emission potential are denoted as $\Gamma$-sub-equation-name (e.g., as $\Gamma_{Langmuir}$, $\Gamma_{Temkin}$ etc).

## 3. Results

### 3.1. Performance of adsorption isotherm equations applied to an environmental soil training and test set

We evaluated the ability of the Langmuir, Freundlich, Temkin and Toth equations to model the exchange of $NH_4^+$ between

adsorbed and aqueous forms. An adsorption curve was characterized for each soil, the data from each soil adsorption experiment was pooled, and then fit using the R nls function. The adsorption curves and fitting parameters are shown in Figure 1 and Table 2. Additionally, as our interest is ultimately in the performance of these adsorption equations at the lower concentration limit, we refit each equation using only the extraction solutions ≤680 mg $L^{-1}$ $NH_4Cl$. The adsorption curves and fitting parameters under those conditions are shown in Figure 2 and Table 3.

160          While the Langmuir (Alnsour, 2020; Guo et al., 2022; Venterea et al., 2015) and Freundlich (Vogeler et al., 2011) equations have been previously reported as being effective at modelling $NH_4^+$ adsorption in soils, we found them to be the least effective of the equations we examined for the full adsorption curves, both over- and under-estimating the adsorbed $NH_4^+$ concentrations (Figure 1), while the Temkin and Toth equations better fit the experimental data. Computing the Akaike information criterion for these equations results in an AIC of -138, -190, -222, and -249 for the Langmuir, Freundlich, Temkin,

and Toth equations, respectively. The absolute value of the AIC is not important, but for a set of models, the model with the lowest AIC is considered the best at fitting the experimental data, indicating an order of Toth>Temkin>Freundlich>Langmuir for model effectiveness. However, fitting only the lower range of the adsorption curves (0 – 40 mM) slightly changed these results, the Toth equation could not be fit by our algorithm, and the resulting AIC values were -93, -157, -152 for the Langmuir, Freundlich, and Temkin equations, respectively, indicating that the Freundlich equation best fit the experimental data. (Note

that AIC values for models fit to different datasets should not be directly compared to one another.)

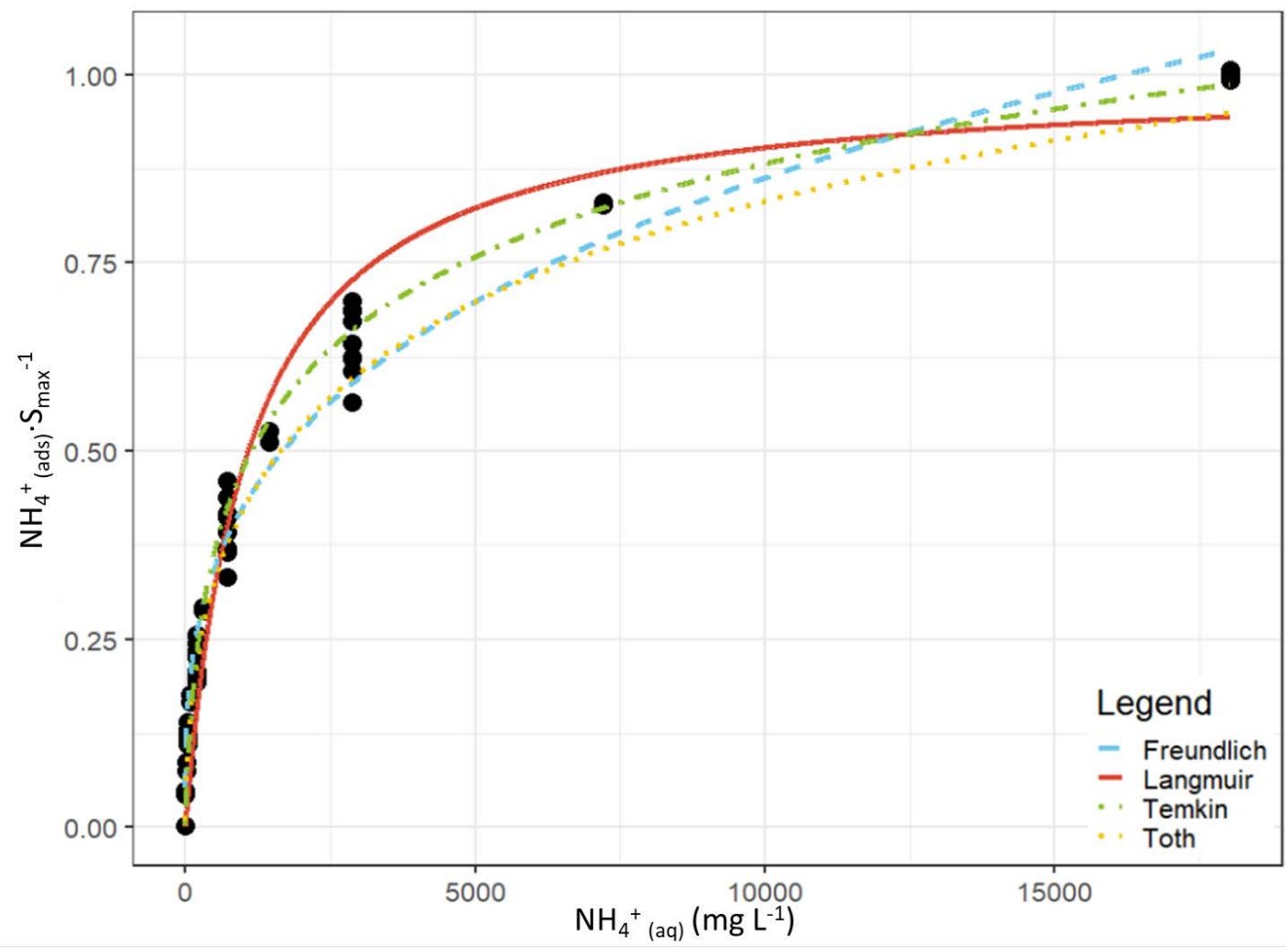

**Figure 1:** Curve-fitting comparison between the Langmuir (red, solid), Freundlich (blue, – – –), Temkin (green, · – · –), and Toth (orange, · · ·) equations. The curves are plotted using the experimental data from all eight soil adsorption experiments, the y-axis is normalized to the maximum adsorption achieved during each experiment.


**Table 2: Comparison of goodness of fit and fitting parameters for the selected adsorption equations**

| Equation | AIC | Parameter 1 | Mean ± Standard error | Parameter 2 | Mean ± Standard error | Exponential Factor |
|---|---|---|---|---|---|---|
| Langmuir | -138 | $K_L$ | $9.29 \cdot 10^{-4} \pm 5.9 \cdot 10^{-5}$ | $S_{max}$ | - | - |
| Freundlich | -190 | $K_F$ | $S_{max}*0.0520 \pm 3.4 \cdot 10^{-3}$ | - | - | $0.3050 \pm 0.0074$ |
| Temkin | -222 | $K_T$ | $1.33 \cdot 10^{-2} \pm 1.2 \cdot 10^{-3}$ | $q_T$ | $S_{max}*0.180 \pm 4.1 \cdot 10^{-3}$ | - |

| | | | | | | |
|---|---|---|---|---|---|---|
| Toth | -249 | $K_{To}$ | $3.10 \pm 0.65$ | b | $S_{max}*2.45 \pm 0.35$ | $0.25 \pm 0.027$ |

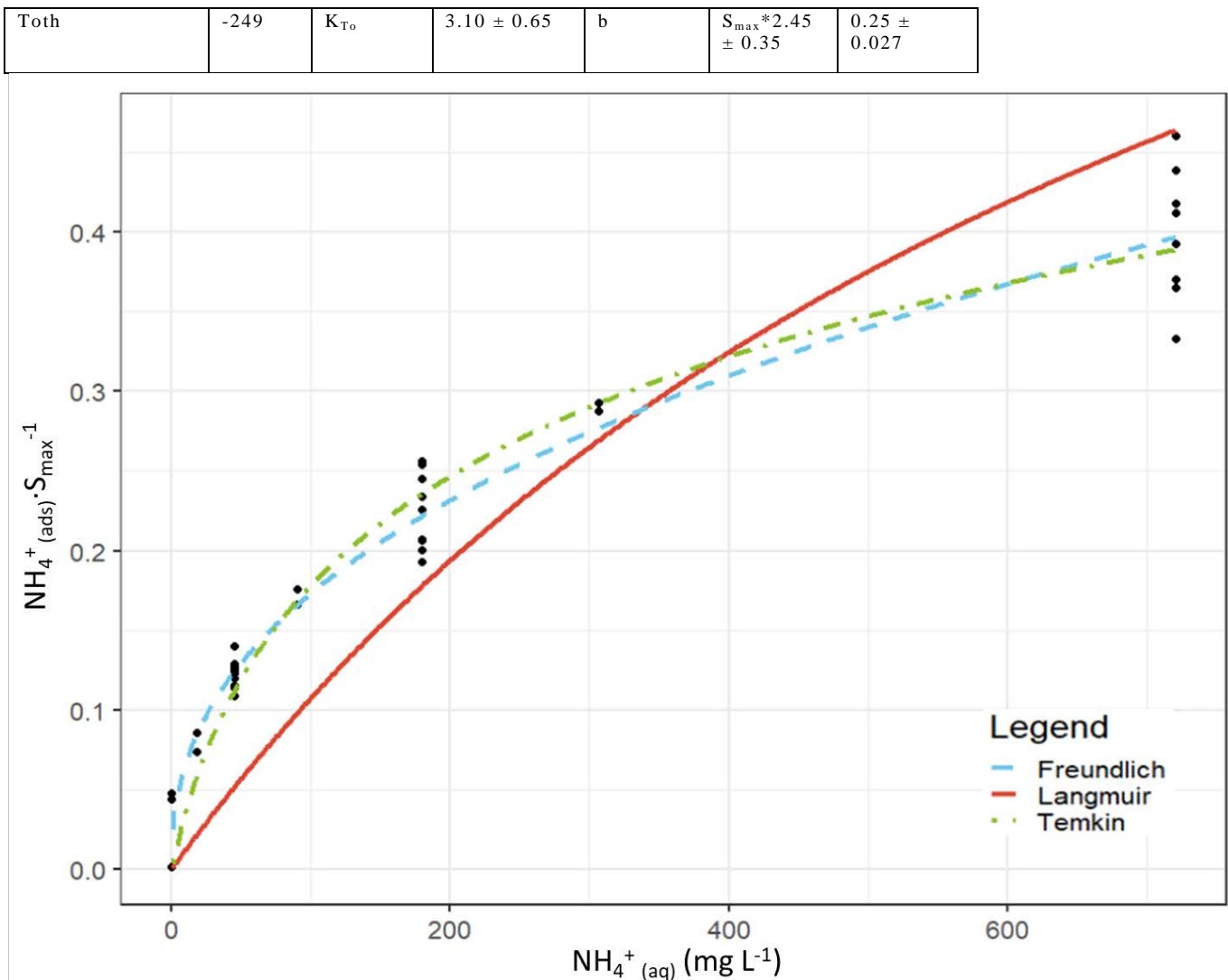

**Figure 2: Curve-fitting comparison between the Langmuir (red, solid), Freundlich (blue, – – –), and Temkin (green, · – · –) equations. The curves are plotted using the experimental data from all eight soil adsorption experiments, but only using the extraction solutions of ≤40 mM NH$_4^+$ for fitting, the y-axis is normalized to the maximum adsorption achieved during each experiment. The Toth equation could not be fit to the experimental data under these conditions.**

**Table 3: Comparison of goodness of fit and fitting parameters for the selected adsorption equations, data fit only for the ≤40 mM NH$_4^+$ solutions**

| Equation | AIC | Parameter 1 | Mean ± Standard error | Parameter 2 | Mean ± Standard error | Exponential Factor |
|---|---|---|---|---|---|---|
| Langmuir | -93 | $K_L$ | $1.2 \cdot 10^{-3} \pm 8.0 \cdot 10^{-4}$ | $S_{max}$ | - | - |

| Freundlich | -157 | $K_F$ | $Smax*0.025 \pm 3.0\cdot10^{-3}$ | - | - | $0.42 \pm 0.019$ |
| Temkin | -152 | $K_T$ | $3.4\cdot10^{-2} \pm 6.5\cdot10^{-3}$ | $q_T$ | $S_{max}*0.120 \pm 8.5\cdot10^{-3}$ | - |
| Toth | Fit did not converge | | | | | |


With our objective being to evaluate how well each equation can fit soils without going through the full characterization procedure, we analyzed the adsorption curves of the low-CEC, medium-CEC and high-CEC soils in our test set using: i) a "typical" CEC of 25, and the fitting parameters from Table 2, ii) the measured CEC and the fitting parameters from Table 2, and iii) by fitting the equations using the least-squares fitting algorithm. Using the first approach, we found that none of the

equations could reasonably fit the experimental data when using an incorrect CEC and that each of the equations fit the experimental data reasonably well using the average parameters and the correct CEC (Figure 3). The relative goodness of fit for each equation was the same for the test set as for the training set, i.e. Toth>Temkin>Freundlich>Langmuir. We also tested an alternative approach for calculating each equation's fitting parameters, in which rather than pooling the normalized data, and then fitting each curve, the curves were fit to each soil adsorption curve separately, and the resulting fitting parameters

were then pooled. A full summary of the alternative fitting parameters and the test-set characterization is given in Figure A1 and Tables A2 – A4.

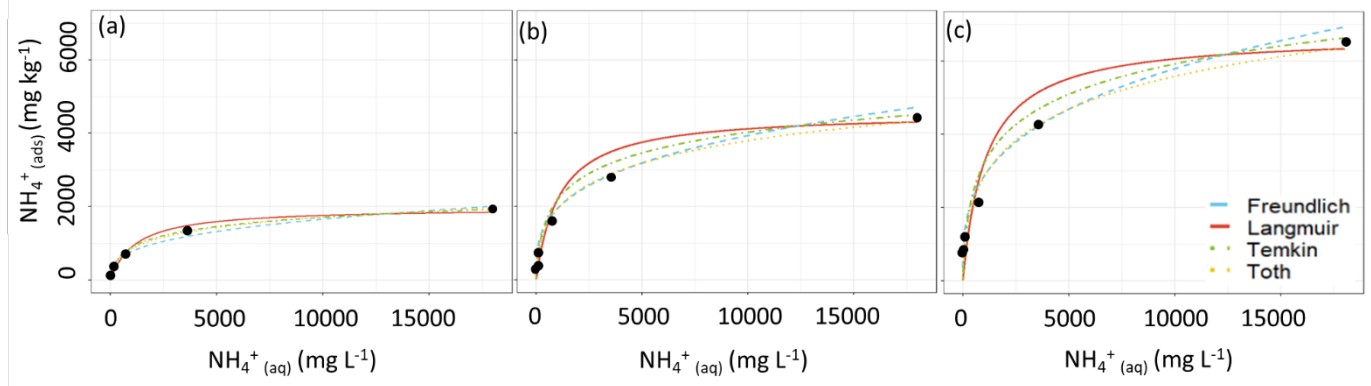

**Figure 3:** Curve-fitting comparison between the unfitted Langmuir (red, solid), Freundlich (blue, – – –), Temkin (green, · – · –), and Toth (orange, · · ·) equations using the fitting parameters from **Table 3** applied to the test set of the a) Low CEC, b) Moderate CEC, c) High CEC samples (method
ii)

Overall, we find that the Toth and Temkin equations best fit the full adsorption curves, while the Freundlich and Temkin equations best fit the low-range adsorption curves. However, both the Freundlich and Toth equations seem to be less robust when applied to this system than the Temkin equation. Firstly, the Freundlich equation, as formulated in Table 2 and Table 3, has an intrinsic dependence on the CEC, which is contrary to the theoretical basis of the Freundlich equation. Using the

alternative fitting approach, the Freundlich equation does not require a CEC-dependence, but in that case, it has high error

when fitting uncharacterized soils. Similarly, for the Toth equation, the alternative approach results in a significantly different set of fitting parameters, which resulted in the Toth equation quite poorly fitting the non-characterized soils. Additionally, while this may be a limitation of the fitting algorithm we selected, we found the Toth equation to be more difficult to consistently fit to our experimental data, with no solution being found for the low-range concentrations. Consequently, taking all of these factors into consideration, we find that the Temkin equation is most suitable for evaluating $NH_4^+$ adsorption from uncharacterized soil samples. For comparison with previous studies, in which the Langmuir equation is the most frequently used adsorption equation, we will continue to analyze it in subsequent sections, despite it performing much worse than the Temkin equation for both the low-range and full-range of data.

### 3.2. Determining aqueous $NH_4^+$ concentrations using the Langmuir and Temkin equations

To relate the total $NH_4^+$ measured ($m_{NH4}$) to the adsorbed portion (S) and the aqueous portion (C), we define Eq.(7):

(7) $m_{NH4} = S + \frac{wC}{\rho}$

Where $m_{NH4}$ is the total mass of $NH_4^+$ per kg soil (mg kg$^{-1}$), w the volumetric moisture content of the soil (L water L$^{-1}$ soil), and $\rho$ is the bulk density of the dry soil (kg L$^{-1}$). By substituting the Langmuir (1) or Temkin (3) equations into Eq.(7), $m_{NH4}$ can be expressed in terms of C:

(8) $m_{NH4} = \frac{S_{max}k_L C}{1+k_L C} + \frac{wC}{\rho}$

(9) $m_{NH4} = q_T ln(1 + K_T C) + \frac{wC}{\rho}$

Where $S_{max}$ is the maximum adsorption capacity (mg kg$^{-1}$); $q_T$ is the product of $S_{max}$ and an empirical fitting constant (mg kg$^{-1}$); and $K_L$ (L mg$^{-1}$) and $K_T$ (L mg$^{-1}$) are empirical fitting constants.

To solve for C, we inverted these equations using *Wolfram Mathematica* (https://www.wolfram.com/mathematica/), yielding Eq.(10) (Langmuir) and Eq.(11) (Temkin).

(10) $C = \frac{-S_{max}\cdot K_L + K_L\cdot m_{NH4} - Z + \sqrt{4K_L\cdot Z\cdot m_{NH4} + (S_{max}\cdot K_L - K_L\cdot m_{NH4} + Z)^2}}{2K_L\cdot Z}$

(11) $C = \frac{-Z + K_T\cdot q_T\cdot W\left(e^{\frac{Z}{K_T\cdot q_T} + \frac{m_{NH4}}{q_T}\cdot Z}\right)}{K_T\cdot Z}$, where **W(x)** is the Lambert W function, and $Z = \frac{w}{\rho}$

If it is assumed that $m_{NH4} \approx S$, then these equations can be simplified to Eq.(12) and Eq.(13) respectively.

(12) $C = \frac{m_{NH4}}{k_L(S_{max} - m_{NH4})}$

(13) $C = \frac{e^{\frac{m_{NH4}}{q_T}} - 1}{K_T}$

We tested this assumption and found that for our unfertilized soil samples, the simplified equations have a positive bias of only 0.57 – 1.5% for the aqueous concentration (C), and that for most soils fertilized with the equivalent of up to 300 kg N ha$^{-1}$, the

simplified equations should have a positive bias of less than 5%. Consequently, we used the simplified equations for our
analysis.

### 3.3.  Determining environmental soil emission potentials

Soil emission potentials were then calculated using the approaches described in Sect. 2.3.3, corresponding to the 'standard' approach, two approaches based on matching the $NH_4^+$ and pH extraction solutions, and two approaches based on applying the Langmuir and Temkin equations to the measured extract concentrations and pH values. The adsorption equations were
calculated using both the low-range and full curve fitting parameters. For the Langmuir calculations, the fit parameters were $K_L = 9.29 \cdot 10^{-4}$ L mg$^{-1}$ (full) and $K_L = 1.20 \cdot 10^{-3}$ L mg$^{-1}$ (low-range), and $S_{max}$ calculated as the equivalent of the CEC in units of mg kg$^{-1}$. For the Temkin calculations, the fit parameters were $K_T = 1.33 \cdot 10^{-2}$ L mg$^{-1}$, and $q_T$ (mg kg$^{-1}$) = 0.180$\cdot S_{max}$ (full) and $K_T = 3.40 \cdot 10^{-2}$ L mg$^{-1}$, and $q_T$ (mg kg$^{-1}$) = 0.120$\cdot S_{max}$. Among the approaches, $\Gamma_{STD}$ results in the highest estimate for the emission potential (17,000 ± 19,000 mol mol$^{-1}$), followed by $\Gamma_{Langmuir}$ and $\Gamma_{SALT}$ (2730 – 3530), and $\Gamma_{Temkin}$ and $\Gamma_{DIW}$ (810 –
1450) (Table 4). Emission potential is linearly related to equilibrium vapor pressure, at a temperature of 15°C, these emission potentials correspond to equilibrium concentrations of 2.4 – 50 ppb. Although CEC is inversely related to the proportion of $NH_4^+$ present in the aqueous phase, the emission potentials increase exponentially as a function of the CEC due to the strong positive correlation between soil pH and CEC (Figure 4).

**Table 4: Comparison of emission potentials determined using the standard, Langmuir, Temkin, salt extraction and DIW extraction methods. Standard deviations represent the underlying variability in the collected soil samples.**

| Method | Equation | Average Emission Potential (mol mol$^{-1}$) (n = 16) | Equilibrium vapor pressure at 15°C (ppb) |
|---|---|---|---|
| $\Gamma_{STD}$ | $\dfrac{[NH_4^+]_{salt}}{[H^+]_{DIW}}$ | 17,000 ± 19,000 | 50 ± 55 |
| $\Gamma_{SALT}$ | $\dfrac{[NH_4^+]_{salt}}{[H^+]_{salt}}$ | 3300 ± 3100 | 9.6 ± 9.0 |
| $\Gamma_{DIW}$ | $\dfrac{[NH_4^+]_{DIW}}{[H^+]_{DIW}}$ | 1450 ± 1900 | 4.2 ± 5.5 |
| $\Gamma_{Temkin}$ (Full) | $\dfrac{e^{\frac{[NH_4^+]_{salt}}{q_T}} - 1}{K_T} \dfrac{1}{[H^+]_{DIW}}$ | 1370 ± 1300 | 4.0 ± 3.8 |
| $\Gamma_{Temkin}$ (Low-range) | | 810 ± 740 | 2.4 ± 2.2 |
| $\Gamma_{Langmuir}$ (Full) | $\dfrac{[NH_4^+]_{salt}}{k_L(S_{max} - [NH_4^+]_{salt})} \dfrac{1}{[H^+]_{DIW}}$ | 3530 ± 3200 | 10 ± 9.3 |
| $\Gamma_{Langmuir}$ (Low-Range) | | 2730 ± 2500 | 7.7 ± 7.3 |

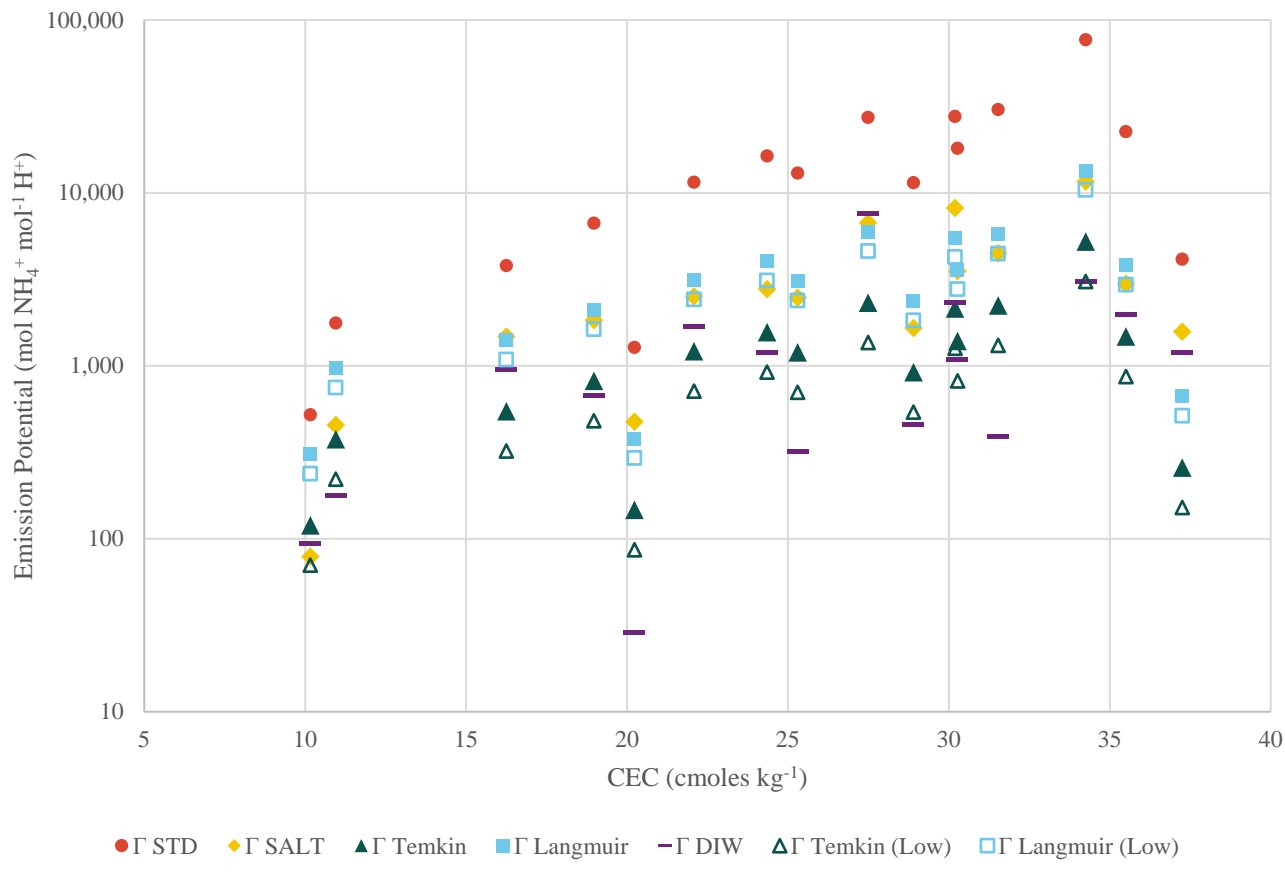

Figure 4: Soil emission potentials using the standard, salt extraction, Temkin, Langmuir and DIW extraction approaches, arranged by CEC.

## 4. Discussion

The conventional approach for calculating emission potentials in soils lacks a solid theoretical foundation, and ought to be applied with caution. Empirically, this approach has performed poorly, with several studies reporting that calculated values of $\Gamma_{STD}$ are unrealistically high and do not correspond well to measured or modelled emission fluxes (Cooter et al., 2010; Flechard et al., 2013; Neftel et al., 1998; Nemitz et al., 2001; Xu et al., 2022). Our assessment that the conventional method overestimates the soil emission potential by at least a factor of 5 is similar to findings reported by Nemitz et al., (2001) and Cooter et al., (2010) who reported needing to reduce their soil emission potentials by a factor of 6.66 and 2 – 3, respectively, for their modelled predictions to match their flux observations. We believe that our empirical treatment of this system with the Langmuir and Temkin equations provides a more reasonable approach to estimate the soil emission potential. Additionally, in our view, the 'like-with-like' extraction approach (e.g. $\Gamma_{SALT}$ or $\Gamma_{DIW}$) has a more mechanistic basis than the conventional approach involving dissimilar extraction solutions. A few recent studies have reported on a similar approach using a Langmuir adsorption

model (Alnsour, 2020; Guo et al., 2022), as well as by matching the $NH_4^+$ and pH extraction solutions (Wu et al., 2023), showing that these methods can be feasibly implemented into an environmental sampling campaign. Venterea et al., (2015) reported that soil $NH_4^+$ partitioning could be effectively modelled using a modified version of the Langmuir isotherm equation of the form $S = \frac{S_{max}C}{K+C}$, where K (mg L$^{-1}$) is the aqueous concentration at which S = ½ S$_{max}$. The Venterea equation is equivalent to the Langmuir isotherm when $K = \frac{1}{K_L}$; for the parameters determined in this study, ½ the saturation capacity is reached at exactly $C = \frac{1}{K_L} = K$, indicating that for our analysis the Langmuir and Venterea equations are equivalent.

How does this approach compare to the implementation of bidirectional exchange of NH₃ in chemical transport models? Widely used models such as GEOS-CHEM and the EPA's CMAQ model soil emission potentials mechanistically, rather than being based on emission potentials prescribed using land-use categories (Bash et al., 2013; Pleim et al., 2013; Pleim, et al., 2019; Zhu et al., 2015). In both models, the exchangeable soil NH₃ has been parameterized as the volumetric molar concentration of $NH_4^+$ in the top 1 – 5 cm divided by the volumetric water content in the soil, or $[NH_4^+] = \frac{NH_4^+ \; mol \; m^{-3}}{\theta \; m^3 m^{-3}}$ (Massad et al., 2010). In this parameterization, the entire soil $NH_4^+$ content of the top layer of soil is treated as being dissolved into the soil water, thus a lower water content would result in a higher concentration of $NH_4^+$, and consequently a higher emission potential. (The parameter used by these models for the resistance to NH₃ emissions from soil is inversely proportional to soil moisture, such that the actual emission of NH₃ from soil would still be proportional to soil water content (Pleim et al., 2013).) In our model of the system, most $NH_4^+$ is present adsorbed to soils, with the aqueous concentration controlled by the equilibrium between adsorbed and aqueous $NH_4^+$, increasing the soil water content would thus allow more $NH_4^+$ to enter the aqueous phase to maintain the equilibrium concentration. (Increased water content could also result in nutrient runoff, reducing the soil $NH_4^+$ content, or increased mineralization, increasing the soil $NH_4^+$ content.) In the subsequent development of the bidirectional model coupled with an agricultural ecosystem model, Pleim et al. (2019) noted that reducing the NH₃ available for emission by implementing the Langmuir adsorption isotherm approach proposed by Venterea et al. (2015) appeared to lead to unexpectedly low fractions of NH₃ available for exchange. It may be that for recently fertilized soils, the applied fertilizer is not in contact with sufficient soil for this equilibrium relation to apply. An updated version of CMAQ (v5.2.1) uses a Langmuir-derived approach described in Venterea et al. (2015) to predict NH₃ bi-directional exchange, and Kelly et al. (2019) explore its indirect impact on PM$_{2.5}$ composition across the U.S., suggesting that the reduced emission potential from our approach is compatible with atmospheric agro-ecosystem modelling.

## 5. Conclusion

This work evaluated the Langmuir, Freundlich, Temkin, and Toth adsorption isotherm equations as applied to environmental soil samples and $NH_4^+$ partitioning. We determined that under ideal conditions the Toth equation was the most effective of these equations at fitting soil adsorption curves, but that the Temkin equation was most effective at predicting the adsorption

behaviour of soil samples with minimal additional characterization required and was effective over both low concentration ranges and the saturation concentration range. Applying this method to a series of environmental soil samples, we determined that the conventional method for directly measuring soil emission potentials may overestimate them by a factor of 5 – 20 (relative to using the Langmuir/Temkin equations respectively). Of the two adsorption equations, the Temkin equation fit the data better, with a significantly lower AIC (-222 vs -138 for the Langmuir equation). Comparing these empirical equations with an alternative approach for determining emission potential based on extracting $NH_4^+$ and pH with the same extraction solutions (i.e. DIW/DIW or Salt/Salt) showed that the Temkin equation fit using the full adsorption range agreed well with the DIW/DIW ratio method but was significantly different when using the low-range fitting parameters.

**Appendix A**

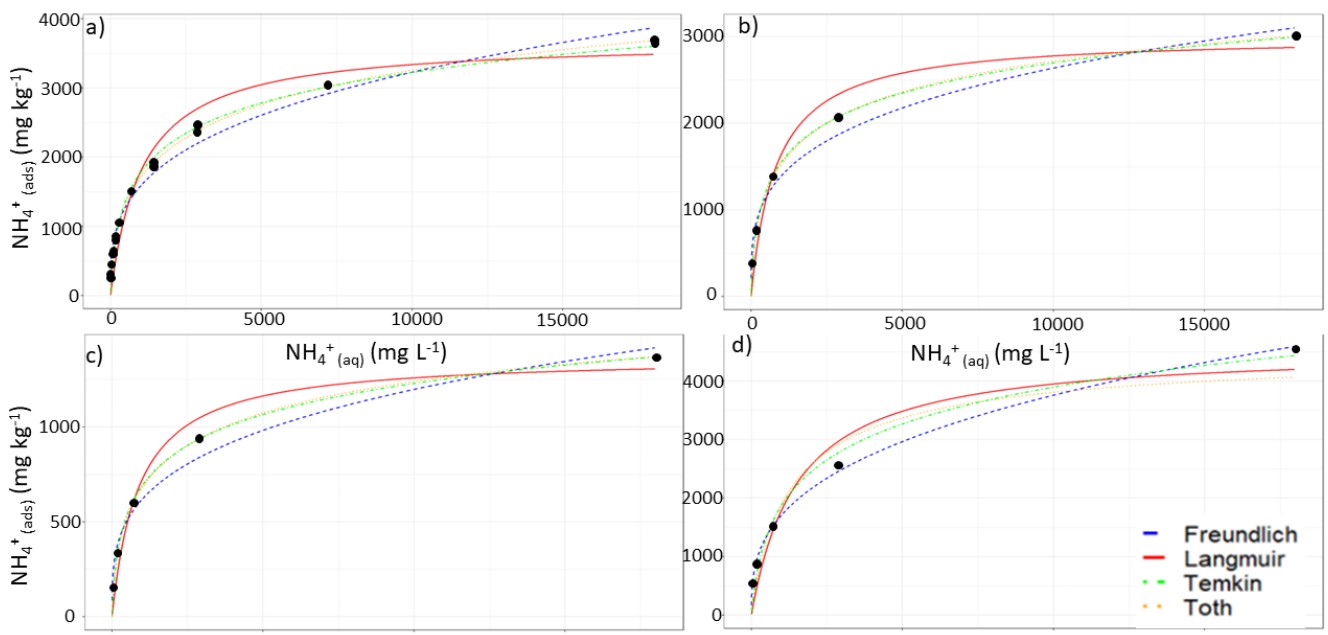

**Figure A1: Curve-fitting comparison between the Langmuir (red, solid), Freundlich (blue, – – –), Temkin (green, · – · –), and Toth (orange, · · ·) equations for soil samples collected from a) King's College Circle, b) the University of Toronto Scarborough forest, c) High Park, and d) Queen's Park.**

**Table A2: Comparison of the residual standard errors, and the range of fitted parameters for adsorption curves fit to the un-pooled data. Parameters given in parentheses are reported for comparison with the parameters in Table 2. Goodness of fit is reported using the residual standard error (RSE) rather than AIC to avoid confusion between ranges of AICs for each equation.**

| Equation | RSE | Parameter 1 | Mean ± sd | Parameter 2 | Mean ± sd | Exponential Factor |
|---|---|---|---|---|---|---|
|  |  |  |  |  |  |  |

| Langmuir | 17.5 – 25.7% | $K_L$ | $9.33 \cdot 10^{-4} \pm 2.0 \cdot 10^{-4}$ | - | - | - |
|---|---|---|---|---|---|---|
| Freundlich | 5.72 – 17.6% | $K_F$ | $197 \pm 60$ <br><br> $(S_{max}*[0.053 \pm 0.012])$ | - | - | $0.3064 \pm 0.023$ |
| Temkin | 3.0 – 16.1% | $K_T$ | $1.34 \cdot 10^{-2} \pm 5.1 \cdot 10^{-3}$ | $q_T$ | $S_{max}*[0.183 \pm 0.013]$ | - |
| Toth | 0.9 – 6.5% | $K_{To}$ | $3.75 \pm 1.3$ | b | $9960 \pm 5590$ <br><br> $(S_{max}*[2.42 \pm 0.72])$ | $0.276 \pm 0.054$ |

315

**Table A3: Residual standard errors when applying average fit parameters from the pooled and (separately fit) training data to the test set soil samples, for i) a CEC of 25; ii) the experimentally measured CEC; iii) when the data is fit using the fitting algorithm**

| Sample Location | CEC (cmole $kg^{-1}$) | $NH_4^+$ (mg $kg^{-1}$) | pH | RSE (%) using estimated fit parameters | | | |
|---|---|---|---|---|---|---|---|
| | | | | Langmuir | Freundlich | Temkin | Toth |
| High Park | 10.95 | 2.906 | 7.04 | 280 (285) | 330 (340) | 320 (330) | 330 (420) |
| | | | | 26 (26) | 22 (26) | 16 (18) | 14 (28) |
| | | | | 25 | 19 | 14 | 8.8 |
| Corktown | 25.3 | 3.552 | 7.82 | 33 (33) | 25 (31) | 22 (26) | 9.5 (39) |
| | | | | 34 (34) | 28 (34) | 25 (28) | 9.7 (43) |
| | | | | 29 | 15 | 15 | 3.8 |
| Riverdale Park East | 37.25 | 2.835 | 7.42 | 64 (63) | 62 (58) | 66 (63) | 84 (62) |
| | | | | 35 (35) | 19 (25) | 29 (31) | 19 (39) |
| | | | | 31 | 11 | 24 | 23 |
| Geometric Mean RSE | Estimated | | | 84 (84) | 80 (85) | 77 (81) | 64 (100) |
| | Estimated with measured CEC | | | 31 (31) | 21 (28) | 21 (25) | 14 (36) |
| | Fitted Equation | | | 28.3 | 14.8 | 17.0 | 9.2 |

**Table A4: Sample information for the adsorption test set**

| Sample | CEC (cmole $kg^{-1}$) | Bulk $NH_4^+$ (mg $kg^{-1}$) | pH | Extractant Solution (mg $L^{-1}$) | Soil Mass (g) | Sum of displaced cations (cmole $kg^{-1}$) | $NH_4^+$ adsorbed (mg $kg^{-1}$) |
|---|---|---|---|---|---|---|---|
| High Park | 10.95 | 2.906 | 7.04 | 18.04 | 1.006 | 0.989 | 178.4 |
| | | | | 36.08 | 1.0124 | 0.949 | 171.2 |
| | | | | 144.32 | 0.9953 | 1.951 | 351.9 |

| | | | | 721.6 | 0.9976 | 4.028 | 726.6 |
|---|---|---|---|---|---|---|---|
| | | | | 3608 | 0.9539 | 7.413 | 1337 |
| | | | | 18040 | 1.0002 | 10.81 | 1950 |
| Corktown | 25.3 | 3.552 | 7.82 | 18.04 | 1.0081 | 1.793 | 323.5 |
| | | | | 36.08 | 1.0095 | 2.222 | 400.9 |
| | | | | 144.32 | 0.9944 | 4.228 | 762.8 |
| | | | | 721.6 | 1.0013 | 9.064 | 1635 |
| | | | | 3608 | 1.0131 | 15.54 | 2804 |
| | | | | 18040 | 1.002 | 24.39 | 4400 |
| Riverdale Park East | 37.25 | 2.835 | 7.42 | 18.04 | 0.9948 | 4.447 | 802.2 |
| | | | | 36.08 | 1.0193 | 4.861 | 876.8 |
| | | | | 144.32 | 1.0151 | 6.690 | 1207 |
| | | | | 721.6 | 1.0304 | 11.79 | 2127 |
| | | | | 3608 | 0.9953 | 23.53 | 4244 |
| | | | | 18040 | 1.027 | 36.26 | 6541 |

**Author contribution statement**

MGD: Investigation, Formal analysis, Writing – Original Draft; KY: Investigation, Methodology; JGM: Supervision, Funding acquisition, Writing – Review & Editing.

**Competing interests statement**

The authors declare that they have no conflict of interest.

## Data Availability Statement

The data used to construct the soil adsorption curves and the measurements of the properties of environmental soils are publicly archived here: https://doi.org/10.5683/SP3/JGRIN0

### Funding statement

This research was supported by a Natural Science and Engineering Research Council (NSERC) Discovery grant and a Grants and Contributions agreement GCXE19S016 with Environment and Climate Change Canada held by Jennifer Murphy. Matthew Davis held a Walter C. Sumner Memorial Fellowship while conducting this research. An Undergraduate Summer Research Award from NSERC supported Kevin Yan during this work.

### Acknowledgements

We thank our colleagues Professor Myrna Simpson and Jenny Oh (University of Toronto) for their helpful discussions. We also thank the University of Toronto ANALEST facility staff for their technical assistance.

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
