# Peer review of "Evaluating adsorption isotherm models for determining the partitioning of ammonium between soil and soil-pore water in environmental soil samples"

_EGUsphere, 2024_

## Author Comment (AC1)

This document summarizes the responses to the feedback from two anonymous reviewers and one community commenter.

**Reviewer 1:**

**General comments**

**The authors present a comparative study on sorption isotherms for ammonium in soils. The manuscript is well-structured and written in a concise style. I see sound scientific work all over, with some minor issues to methodological aspects. But I am sure this could be sorted out...**

We thank the reviewer for their useful feedback and provide responses to their specific comments below.

**Specific comments**

**15 'conventional' I suggest to be a bit more precise at this point**

Changed: "calculated using a conventional nutrient analysis method" to:

"determined using a conventional high-salt extraction procedure to determine the soil ammonium content"

**26 many references of NH3 emissions from plants not from soil**

Thank you for the feedback, our intention was to give citations on the development and application of the ammonia bidirectional exchange model generally, most of the early work did focus on plants, we have added several more recent references that apply the model to soil as well, including:

Guo, X., Pan, D., Daly, R. W., Chen, X., Walker, J. T., Tao, L., McSpiritt, J., & Zondlo, M. A. (2022a). Spatial heterogeneity of ammonia fluxes in a deciduous forest and adjacent grassland. *Agricultural and Forest Meteorology*, *326*(July), 109128. https://doi.org/10.1016/j.agrformet.2022.109128

Walker, J. T., Chen, X., Wu, Z., Schwede, D., Daly, R., Djurkovic, A., Oishi, A. C., Edgerton, E., Bash, J., Knoepp, J., Puchalski, M., Iiames, J., & Miniat, C. F. (2023). Atmospheric deposition of reactive nitrogen to a deciduous forest in the southern Appalachian Mountains. *Biogeosciences*, *20*(5), 971–995. https://doi.org/10.5194/bg-20-971-2023

Wentworth, G. R., Murphy, J. G., Gregoire, P. K., Cheyne, C. A. L., Tevlin, A. G., & Hems, R. (2014). Soil-atmosphere exchange of ammonia in a non-fertilized grassland: Measured emission potentials and inferred fluxes. *Biogeosciences*, *11*(20), 5675–5686. https://doi.org/10.5194/bg-11-5675-2014

Zhang, L., Wright, L. P., & Asman, W. A. H. (2010). Bi-directional air-surface exchange of atmospheric ammonia: A review of measurements and a development of a big-leaf model for applications in regional-scale air-quality models. *Journal of Geophysical Research Atmospheres*, *115*(20). https://doi.org/10.1029/2009JD013589

**29-31 I suggest to delete this motivation for sampling natural soils. The sorption mechanism (or the physico-chemistry) of soils with N fertilization, and probably higher NH4 and NH3 concentrations, is exactly the same as for natural soils. The same sorption isotherms are applied...**

We agree with the reviewer on this point; however, in the atmospheric chemistry modelling field, ammonia sorption in both managed and unmanaged soils is not always consistently addressed, perhaps because the focus is generally on recently fertilized soils where a sorption equilibrium may not have been reached. We have added some clarifying language to this section:

"Consequently, ammonia volatilization models may parameterize all or most of the soil ammonia as being readily able to exchange with the atmosphere, which may be reasonable for recently fertilized soils, but not for unmanaged soils"

**43 '... soil sample able to participate ...' I suggest to rewrite**

Changed:

"In this manuscript, we explore a variety of adsorption isotherm models with the goal of identifying a simple approach to relate the quantity of ammonium in a soil sample able to participate in bidirectional exchange with the total amount of ammonium, as a function of other readily measureable quantities."

To:

"In this manuscript, we explore a variety of adsorption isotherm models with the goal of identifying a simple approach to relate the total quantity of ammonium in soil to the aqueous fraction of ammonium that can participate in bidirectional exchange with the atmosphere."

**110 'RSE' I strongly suggest to compute the Akaike information criterion (AIC) or the Baysian information criterion (BIC) instead of RSE for model identification. AIC relates the model error to the number of parameters.**

We appreciate the feedback and have replaced the use of the RSE with the Akaike information criterion throughout the manuscript. This does not qualitatively alter our findings (as the rank order of goodness-of-fit is the same, whether determined by AIC or relative RSE), but we appreciate the suggestion of a more widely accepted metric.

**112 parameters of non-linear functions usually do not average well, and usually using the geometric mean is better than using the arithmetic mean**

Thank you for this comment, we originally calculated these values using the geometric mean, but we chose to report the arithmetic mean because we were not confident that the geometric mean would be accepted as a metric, and the geometric mean and arithmetic mean were quite similar for our dataset. As per your comment on line 138, we think a more straightforward way to approach this section would be to only report on the pooled data, rather than by comparing the pooled data with the averaged parameters.

**117 please justify the selection of min, max and some mean value instead of a random sample. And a higher number than just 3 samples in the test data set would actually be required**

Added the following language to explain the selection:

As the original training set mostly consisted of samples with CECs from 20 – 30 (with two samples with CECs of 7.6 and 16), we chose two samples that were significantly different than the average training set sample, as well as one similar sample to determine whether the fitting parameters could be used for 'extreme' samples, or only for samples similar to the training set.

**119-120 points 'i.' and 'ii.'; do you intend to test the potential of CEC to predict sorption?! That's a good idea anyways, but it would be nice to mention it already in the introduction as a goal.**

We have added language in the introduction (at line ~55) to make our intention more clear:

"…each of these equations incorporate a saturation point or maximum adsorption capacity ($S_{max}$, mg kg$^{-1}$), in this work, we investigate the potential to calculate these adsorption equations as a function of the measured CEC (converted to mg of $NH_4^+$ kg$^{-1}$soil), rather than treating $S_{max}$ as a calculated fitting parameter.

**138 pooling is theoretically the more appropriate procedure**

As per the response to the comment for line 112, we have removed the discussion on the separate fit + averaged approach, and will primarily report the 'pooled data' approach.

**148 'individual' instead of 'individually'?**

As part of the response for 138, this line has been deleted.

**150 For the figure caption I suggest to replace 'standardizing the y-axis' with 'normalized'**

Thank you, changed.

**Table 2 Use either 'SE' or 'Standard error' in the column headers for consistency; I guess the number of parameters stated for Langmuir is wrong, this is also =2; For Freundlich I would move 'Smax*0.052...' one column to the right?!**

SE changed to standard error for consistency. For the number of parameters in the Langmuir model, we are only fitting one parameter, Smax is treated as a measured parameter. For the Freundlich entry, while our definition of $K_F$ is more similar to the definitions for parameter 2 in the other equations, the Freundlich equation only has 1 non-exponential parameter, so keeping it in the first column seems best to us.

**161 would be nice to have a sort of pedotransfer function to estimate Smax or other sorption parameters from CEC. I think even the small data set present in this study could be used for an initial test ...**

To clarify, the approach used in this study was indeed to treat the CEC as the saturation capacity ($S_{max}$) for $NH_4^+$; the CEC has been previously reported to be a good soil characteristic for estimating ammonia sorption in soil, e.g. Vogeler et al., 2011

Vogeler, I., Cichota, R., Snow, V. O., Dutton, T., & Daly, B. (2011). Pedotransfer Functions for Estimating Ammonium Adsorption in Soils. *Soil Science Society of America Journal*, *75*(1), 324–331. https://doi.org/10.2136/sssaj2010.0192

**162 well, there are differences in the model errors, however for field conditions, e.g in the context of agroecosystem models, it actually does not matter much. Simply because the sorption if NH4 is very, very high. And it does not really matter that much how high Smax is as long as the sorption at low concentrations near zero is estimated well. And in that range of concentrations I assume the four isotherms provide very similar results. Something that should probably be picked up in the discussion.**

We did calculate the low concentration behavior of each of the equations, and found that (for our fitting parameters) the Temkin, Langmuir and Toth equations performed relatively similarly (at reasonable soil concentrations of ammonium), while the Freundlich equation predicted what we considered to be unrealistically high sorption to soils, using the full-range fitted parameters. This section has been edited following a suggestion from commenter CC1, but in general, we chose to exclude the Freundlich and Toth equations from consideration due to practical concerns with applying those functions to our experimental dataset, and we feel additional discussion of those functions in the discussion is not warranted.

**167 'an additional parameter' this is where the AIC is helpful...**

This line has been removed due to the edits for CC1's suggestions. However, to clarify, we did not mean that the Toth equation was trivially superior at fitting the experimental data because it had an additional degree of freedom which tends to always result in an increased goodness of fit, which is one of the situations that the AIC is used to detect. We meant instead that we felt that it was over-fitting our experimental data, resulting in a non-trivial increase in goodness of fit, but was not generalizable.

**155-160 I think this paragraph would benefit from additional references to literature**

Due to the edits for CC1's suggestions, this paragraph has ben removed.

**figure 2 I think the label of the y-axis is wrong. This probably is the non-normalized sorbed NH4 concentration, otherwise it should also vary around values <1 ...**

Thank you for catching this oversight, the manuscript has been corrected.

**190-194 I am not sure I understand the idea behind Eqs. 9 to 13. Eq. 7 could be rearranged to C=((mNH4-S)*p)/w and S can be computed from any of the isotherm equations given in Table 1. OK, then you need two equations, but this is easy to compute. For me it is particularly difficult to understand the assumption wC/p=0. You basically neglect the liquid phase concentration of NH4, and this is basically not required. It just makes your estimate of C worse.**

The problem with the suggested approach is that the soil nutrient assay used to measure ammonium in soil measures neither S nor C, but only the total quantity, mNH4. Perhaps it might be possible to devise an approach that would do so, but our objective is to require as few additional analysis steps that are not commonly used for soil measurements as possible. Solving the problem algebraically is, in our view, a better approach.

As for the simplifying assumption that $wC/p \approx 0$, we agree that it is not a necessary assumption, but our opinion is that it is a helpful simplification, and we have presented both the approximate and exact solutions so that a prospective reader can use whichever equation might be more appropriate for their needs.

**196 yes, that 0.57-1.5% is probably the ratio of C/S, and just a consequence of neglecting the liquid phase NH4 concentration**

That is correct. Note that (in Table 4) the coefficient of variation in our environmental soil samples (representing real variation in the soil ammonium concentrations across the city) is on the order of ±100%, a positive bias of ~1% is not significant.

**226 replace 'more sound' with 'mechanistic'?**

Replaced (~line 226):

"a more sound theoretical basis"

With:

"a more mechanistic basis"

**251-252 please remove the phrase '; and more research ... of both approaches.'**

Removed

**253-255 please delete or relate to any of your results**

Added clarifying language to emphasise the connection with our results:

"...suggesting that the reduced emission potential from our approach is compatible with atmospheric agro-ecosystem modelling"

**264 how much better?  % ...**

Edited this section to refer to the AIC as a criterion rather than the RSE, as a result of the other edits suggested, we have now made it more clear that the Temkin model is significantly better than the Langmuir model, and that we are only presenting both models so as to compare with previous reports in the literature that use the Langmuir model.

**Reviewer 2:**

**In Section 2.2.2. Adsorption curve characterization (Line 85), if I understand correctly, an indirect method used to quantify adsorbed ammonium and amount remaining in solution based on the displaced Na+, Mg2+, Ca2+ and K+ ions measured in solution. Could not the ammonium remaining in solution been measured directly and then used to determine adsorbed amount by subtraction, perhaps to confirm with the indirect measurements of other cations? Do the authors believe that the indirect measurement is more reliable than direct for some reason?**

The reviewer is correct in their interpretation of our procedure, and are also correct that the ammonium remaining in solution could be measured directly to determine the adsorbed quantity by subtraction. However, we felt that the analytical uncertainty was too high for this to be an effective approach. The difference between the prior and posterior solution concentrations is on the order of 1 – 2 %, but because the solution concentrations are outside the calibrated linear range of our instrumentation (initially we were using ion chromatography to quantify NH4+), they would have to be diluted after the adsorption trials. We felt that the propagated errors throughout this procedure would make it too difficult to reliably detect a 1 – 2 % concentration difference.

**Community commenter (John Walker):**

**This paper tackles the important question of how to properly estimate the soil NH3 compensation point for air-surface exchange modeling.  To this end, the authors present an extensive and much needed examination of soil NH4+ sorption characteristics.  I have a few questions about the methodology that might be considered and a suggestion to improve the usability of the results.**

We appreciate this useful feedback and provide responses to the specific comments below.

**Line 86:  Typically, the quantity of NH4+ adsorbed onto the soil is inferred from a measurement of NH4+aq in the equilibrated solution (see the papers referenced in Section 1.3).  In this study, the quantity of sorbed NH4+ is inferred from measurements of Na+,**

**Mg2+, Ca2+ and K+ in solution. A justification of this approach would be helpful. Was a comparison conducted to see if the two approaches yield similar sorption parameters? If so, it would be a useful addition to the paper.**

To clarify (for the editors and others), the reference procedure from Venterea et al. 2015 is to add 5, 10, 50, etc. mg/L of $NH_4Cl$ to separate masses of soils, and to then measure the difference between the post-equilibration solution and the original concentration, taking into account the natural ammonium present in the soil. We did try to follow this procedure as written, however, in our preliminary work we found that a higher concentration range was needed to saturate the soil adsorption capacity than was used by Venterea et al. Moreover, we originally were using ion chromatography as our analysis method, and found that for our equipment (Dionex ICS-2000, CS-17 column, 100 µL sample loop), concentrations of $NH4+$ greater than ~5 mg/L were outside the linear calibration range, requiring samples to be diluted to be quantified. However, the analytical uncertainty (calculated with a propagation of errors analysis) was then too high for us to feel confident that this method could be used to detect the expected concentration change. The alternative approach based on the cation-sum displacement method was convenient experimentally, and in our conceptualization of this system where the CEC is a key parameter for understanding the adsorption of ammonium to soils, adapting a CEC-measurement technique to measure ammonium adsorption was logical. We have added clarifying language to line 86:

In the cation sum method, the total quantity of adsorbable cations is determined by saturating the soil with an index cation (in this procedure, and generally, $NH_4Cl$), which replaces the displaced cations in the soil's adsorption sites, thus the displaced cations measured for each $NH_4Cl$ solution is indicative of the quantity of added $NH_4^+$ adsorbing onto the soil.

**Paragraph beginning line 143: How do the different equations compare over a narrower range of NH4+aq? The maximum NH4+aq concentration in this study (Figure 1) is larger than in the references cited (Alnsour, 2019; Venterea et al., 2015; Vogeler et al., 2011). This comparison could be informative to the applicability of the models to soils with low NH4+ concentrations.**

This was a helpful suggestion, we recalculated the fit over a narrower range of concentrations (comparable to the references above), and found that the Freundlich and Temkin equations performed well at the lower concentration range. We think that using the Freundlich equation for this analysis is less desirable because we needed to include the CEC/Smax as a parameter in the Freundlich equation for it to be generalizable between soils, which is inconsistent with the theoretical basis of the Freundlich equation. We have added the "low-range" version of the equations throughout in the revised manuscript, with too many changes to list here directly.

**Line 192. I would encourage the authors to avoid simplifying equation (7) to exclude the aqueous portion. Understandably, it is a small mass compared to the sorbed portion but the simplification is unnecessary.**

We agree with the reviewer that the simplification is unnecessary, consequently we have provided the full version of both equations as well as the simplified versions. For our analysis, where the underlying variability in our environmental samples is on the order of ±100%, we do not consider a positive bias of ~1% to be significant.

**General comment:  I encourage the authors to include a table summarizing the fitting (sorption) parameters for each equation for each sorption experiment, grouped in a way that they can be related to the corresponding basic soil chemical parameters (CEC, pH, extractable NH4+).  A more complete and detailed summary of the data will make the results more widely usable.**

Unfortunately, we did not measure the original extractable NH4+ for the soils that we used to derive the fitting parameters for the sorption experiments (training set), but only for the test set. Following this suggestion, we have added Table A4 to the manuscript, which provides the experimental data from the adsorption isotherm trials with the test set.

---

## Author Response (AR2)

AR1

Simplifying Equation 7, well, the community commenter and I did not like the idea of neglecting the liquid phase concentration here. The reason is that the emissions of Ammonia occurs as a consequence of the liquid phase concentration of NH4 (Henries law...). And also the crop uptake of ammonium occurs via the liquid phase NH4. I understand that mathematically this is a small fraction of the total ammonium in the soil. However, it is the relevant fraction. Of course you can publish both options, but I think in the soil science community this simplification will not be accepted.

*I think our previous responses misunderstood the critique in this section, we are not neglecting the aqueous phase concentration, which is the quantity that is calculated using equations 7 – 13. The simplified approach is merely a numerical approximation for equations 10 - 11 which results in a typical approximation error of <1.5% in the value of the aqueous phase concentration. We have edited lines 218 and 222 to clarify our intention.*

'number of parameters' I hope this does not sound too picky. The freundlich and Langmuir equations both have two parameters. No matter how one of the parameters was derived. Mathematically they have two parameters and even in table 1 the units of two parameters are given for freundlich and Langmuir...

*Table 2, Table 3 and Table A2 have been edited to remove the column indicating the number of parameters. (Note that Microsoft word's track changes function does not track the removal of columns from tables).*

*Other changes:*

*Several minor typographical and grammar changes have been made, highlighted in track changes.*

---

## Author Response (AR3)

Response to reviewers:

In general, all technical and editorial suggestions were accepted.

*Briefly explain to the reader why you have used two different solutions (101)*

The various references to using two solutions in this paragraph were removed, and a separate paragraph was added below (now on lines 112-117):

> "Conventionally, the soil emission potential is calculated as a function of the soil pH ($H^+_{DIW}$) and salt solution-extracted $NH_4^+$ ($NH_4^+{}_{SALT}$). In addition to investigating whether adsorption isotherm equations could be applied to estimate $NH_4^+{}_{(aq)}$ from the total soil $NH_4^+$, we investigated the impact of calculating the emission potential using a 'like-with-like' ratio of $H^+_{DIW}$ with $NH_4^+$ extracted using DIW as the solvent, as well as the ratio of $NH_4^+{}_{SALT}$ to $H^+$ determined from a salt-solution:soil slurry. Consequently, the pH was also determined as described above, but using a 0.01 M $CaCl_2$ solution in place of DIW, while soil $NH_4^+$ was also determined as described above, but using DIW as the solvent."

*I don't understand why not all 16 soil samples were used as a test set. (122)*

The 16 environmental soil samples were collected to measure the impact of the different approaches on the soil emission potential, a subset were used to validate the parameters calculated using the 8 fully characterized soil samples. Fully characterizing soil samples (which was done for the 8 + 3 soil samples) is labor-intensive, and characterizing the additional 13 samples was not considered a good use of resources.

*Why is there no +/- se for Smax? (Table 2, Table 3)*

These tables show the calculated fitting parameters and errors for the investigated adsorption equations, Smax was directly measured and does not have an associated fitting error.

*Define emission potential and add unit (233)*

An explanation of the emission potential was added to the introduction, where it was missing (in addition to the definition that was already given in the method section)

> The propensity for ammonia to volatilize from liquid reservoirs is parameterized by the emission potential ($\Gamma$), which is calculated as the ratio of aqueous $NH_4^+$ to $H^+$.